# Anatomical and functional organization of the human substantia nigra and its connections

**Yu Zhang, Kevin Michel-Herve Larcher, Bratislav Misic, Alain Dagher\***

Montreal Neurological Institute, McGill University, Montreal, Canada

**Abstract** We investigated the anatomical and functional organization of the human substantia nigra (SN) using diffusion and functional MRI data from the Human Connectome Project. We identified a tripartite connectivity-based parcellation of SN with a limbic, cognitive, motor arrangement. The medial SN connects with limbic striatal and cortical regions and encodes value (greater response to monetary wins than losses during fMRI), while the ventral SN connects with associative regions of cortex and striatum and encodes salience (equal response to wins and losses). The lateral SN connects with somatomotor regions of striatum and cortex and also encodes salience. Behavioral measures from delay discounting and flanker tasks supported a role for the value-coding medial SN network in decisional impulsivity, while the salience-coding ventral SN network was associated with motor impulsivity. In sum, there is anatomical and functional heterogeneity of human SN, which underpins value versus salience coding, and impulsive choice versus impulsive action.

DOI: https://doi.org/10.7554/eLife.26653.001

## Introduction

Dopamine innervation to the cerebral hemispheres originates in the substantia nigra (SN) and ventral tegmental area (VTA) of the midbrain. In monkeys, SN/VTA dopamine neurons display variations in both anatomy and function. SN neurons can be divided into three tiers based on their staining, morphology, and connectivity with the striatum (*Haber, 2014*; *Haber and Knutson, 2010*): moving from a dorso-medial to ventro-lateral location in midbrain, dopamine neurons project to limbic, associative and then motor striatum. All three subdivisions send dendrites ventrally into the adjacent SN pars reticulata (*Haber and Knutson, 2010*). Distinct functional characteristics have also been reported for SN/VTA neurons by recording neural activity during appetitive and aversive outcomes (*Matsumoto and Hikosaka, 2009*; *Nomoto et al., 2010*). Cells in ventromedial SNc and VTA encode a value signal, being excited by appetitive events and inhibited by aversive events. Neurons in lateral SN may encode a salience signal, responding equally to appetitive and aversive stimuli.

Dopamine plays a crucial role in decision-making and reinforcement learning by encoding a reward-prediction error signal (*Bayer and Glimcher, 2005*; *Glimcher, 2011*; *Schultz et al., 1997*). More recently, the role of dopamine in motivational and cognitive processing has been extended by descriptions of responses not only to rewarding outcomes but also to novel, salient, and possibly aversive experiences (*Bromberg-Martin et al., 2010*; *Lisman and Grace, 2005*; *Matsumoto and Hikosaka, 2009*; *Redgrave and Gurney, 2006*).

An important clinical aspect of dopamine signaling is its role in impulsivity, defined as a tendency to act rapidly and prematurely without appropriate foresight (*Dagher and Robbins, 2009*; *Dalley and Robbins, 2017*; *Morris and Voon, 2016*). Impulsivity is a key feature of drug addiction, obesity, and attention deficit hyperactivity disorder (ADHD). It can be divided into different components (*Meda et al., 2009*). Decisional impulsivity is characterized by a tendency to make maladaptive

**\*For correspondence:**
alain.dagher@mcgill.ca

**Competing interests:** The authors declare that no competing interests exist.

or inappropriate choices and is typically tested with the Delay Discounting task. Motor impulsivity, on the other hand, refers to premature responding or an inability to inhibit an inappropriate action, and can be tested using Go/No Go type tasks. Discrete neural networks may underlie different forms of impulsivity: VS and ventromedial prefrontal cortex (vmPFC), which encode stimulus value, are implicated in decisional impulsivity (*Kable and Glimcher, 2007*; *McClure et al., 2004*; *Sellitto et al., 2010*); somatomotor cortex, supplementary motor area (SMA), inferior frontal gyrus (IFG), anterior insula and dorsal striatum are thought to play a role in motor impulsivity (*Bari and Robbins, 2013*; *Cai et al., 2014*; *Chikazoe et al., 2009*). All these brain regions are interconnected with SN: they receive dopamine innervation and send back direct or indirect projections that can modulate dopamine neuron activity (*Haber and Knutson, 2010*).

In this study, we sought to determine anatomical and functional subdivisions of human SN, their connections with striatal and cortical regions, and their role in value versus salience coding, and in different forms of impulsivity. To date, the in-vivo mapping of SN connectivity in humans has been challenging. The brainstem is prone to artifacts from head movement, pulse and respiration, as well as image distortions during data acquisition. Besides, the SN is a relatively small structure that connects with cortical and striatal regions through dense tracts in the internal capsule (*Meola et al., 2016*), which causes difficulties for diffusion tractography (*Jbabdi et al., 2015*). Here, we attempt to overcome these limitations using data from the human connectome project (HCP), taking advantage of the high spatial resolution and rich collection of multimodal measures. First, we used diffusion MRI (dMRI) to identify subdivisions of SN according to their connectivity patterns with the rest of brain. We then mapped the distinct connectivity profiles for each subdivision. Next, we used an fMRI gambling task to differentiate Blood Oxygen Level Dependent (BOLD) activity in SN subdivisions and their projections in terms of responding predominantly to value or salience. Finally, we related individual differences in value and salience coding during the gambling task to measures of decisional and motor impulsivity to reveal dissociable neural substrates underlying impulsive choice and impulsive action.

## Results

### Subdivisions of substantia nigra

The SN was parcellated based on spectral clustering of the connectivity patterns of SN voxels (*Figure 1*) as described previously (*Fan et al., 2016*). Three stable subdivisions were identified in the SN of each hemisphere (*Figure 2B*): a dorsolateral area corresponding to lateral part of SN pars compacta (lateral SNc – hereafter: lSNc), a dorsomedial area corresponding to medial part of SN pars compacta (medial SNc - mSNc) and a ventral area (vSN). The Montreal Neurological Institute (MNI) coordinates for the center of mass of each SN subdivision are listed in *Table 1*. The three subdivisions were of similar volume on average (vSN: 93/90 $mm^3$ for left/right; mSNc: 90/117 $mm^3$ for left/right; lSNc: 117/98 $mm^3$ for left/right). This separation of dorsomedial, lateral and ventral SN coincides with descriptions of dorsal, middle and ventral tiers of midbrain dopamine cells in primates (*Figure 2—figure supplement 1*), which have distinct afferent and efferent striatal and cortical projections (*Haber, 2014*; *Haber and Knutson, 2010*).

The optimum number of subregions within SN was determined by evaluating both reproducibility of parcellation using repeated split-half resampling and topological similarity across hemispheres. As shown in *Figure 2D*, the three-subdivision parcellation of SN showed both high reproducibility (mean normalized mutual information (NMI) = 0.85 and 0.88, respectively, for left and right SN) and high inter-hemispheric topological similarity (mean NMI = 0.68). The same conclusion was drawn from the other stability indices, the Dice coefficient and Cramer's V (*Figure 2—figure supplement 3*. The stability of the parcellation is also supported by the probabilistic maps of each subdivision (*Figure 2C*), where the intensity represents the probability of subdivision assignment over the population at each SN voxel. Finally, the replication of the parcellation procedure on a second group of 60 randomly selected subjects from the HCP dataset yielded very similar results (NMI = 0.95; *Figure 2—figure supplement 2*).

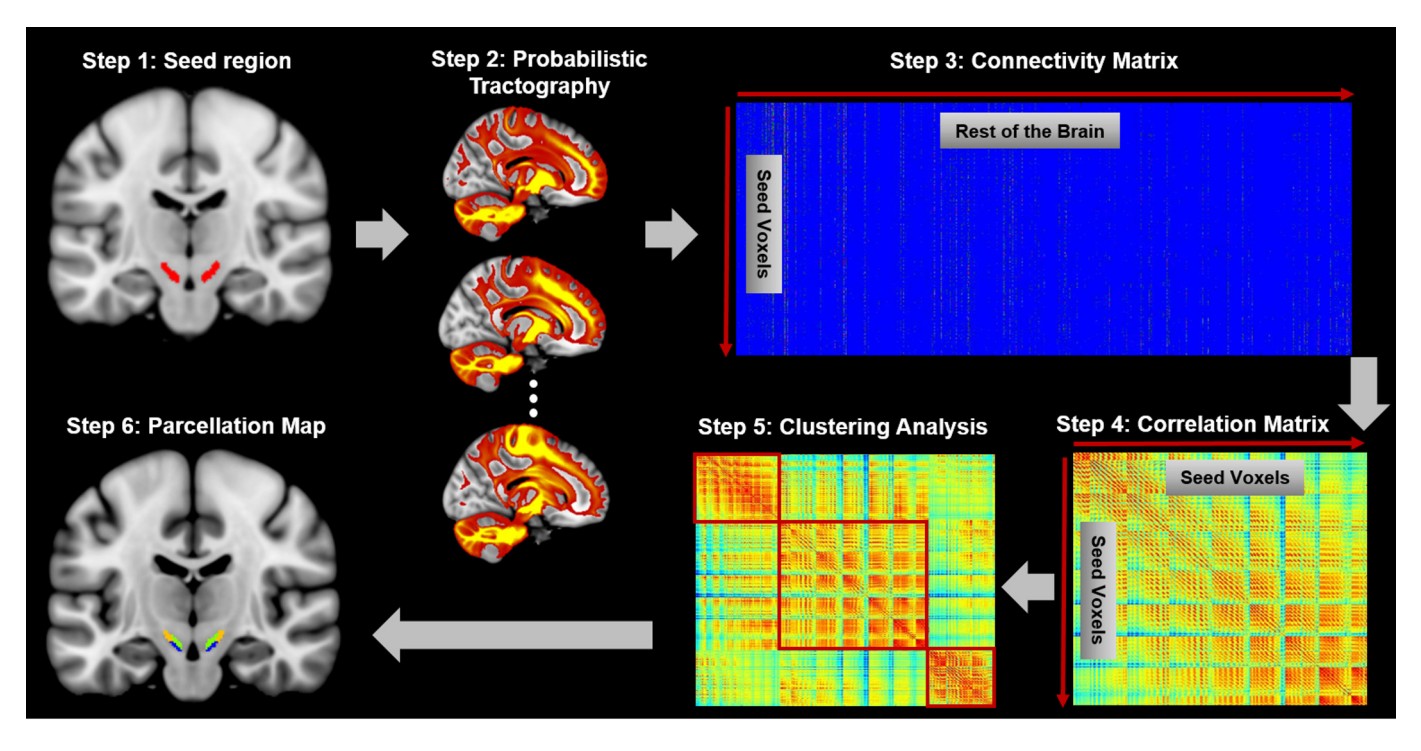

**Figure 1.** Connectivity-based brain parcellation procedure. After defining the seed region (step 1), probabilistic tractography was applied by sampling 5000 streamlines at each voxel within the seed mask (step 2). Whole-brain connectivity profiles were used to generate a connectivity matrix with each row representing the connectivity profile of each seed voxel (step 3). Next, a correlation matrix was calculated as a measure of similarity between seed voxels (step 4). Then, spectral clustering was applied to the similarity matrix (step 5) and multiple subdivisions were identified within the seed region (step 6). The entire procedure was applied independently for each hemisphere and each subject.

DOI: https://doi.org/10.7554/eLife.26653.002

The following figure supplements are available for figure 1:

**Figure supplement 1.** Multi-slice view of the SN seed mask on group averaged brains.

DOI: https://doi.org/10.7554/eLife.26653.003

**Figure supplement 2.** Spatial locations of the four regions of interest.

DOI: https://doi.org/10.7554/eLife.26653.004

## Connectivity patterns of SN subdivisions

Distinct connectivity profiles were identified for each subdivision of SN by performing probabilistic fiber tractography from each subdivision (*Figure 3*) on 430 HCP datasets. Specifically, the dorsolateral subregion (i.e. lateral SNc) mainly connected with the somatic motor and sensory cortex in pre-/post-central gryus; the dorsomedial subregion (i.e. medial SNc) showed dominant connections with limbic regions including lateral and medial OFC, hippocampus and amygdala. The ventral subregion (i.e. vSN) preferentially connected with prefrontal cortex, anterior cingulate cortex and anterior insula. These connectivity maps reveal a limbic-cognitive-motor organizational topography of SN fiber projections. They also demonstrate a rotation in topology from medio-lateral in SN to ventro-dorsal (and rostro-caudal) in cortex.

This limbic-cognitive-motor topology of SN projections was also evident from the maximum probability map (MPM) of the tractograms (*Figure 4* and *Figure 4—figure supplement 1*). Particularly, in prefrontal cortex, a clear rosto-caudal pattern of SN projections was present, with medial SNc mainly connecting with the most rostral and ventral part including OFC and frontal pole, vSN connecting to lateral and dorsomedial prefrontal cortex including middle frontal gyrus and inferior frontal gyrus, and lateral SNc showing connections with the sensorimotor and somatosensory cortex. A similar rostro-caudal distribution of fiber tracts was also seen in striatum, with medial SNc mainly connecting with ventral striatum, vSN connecting via the anterior limb of the internal capsule with the body of

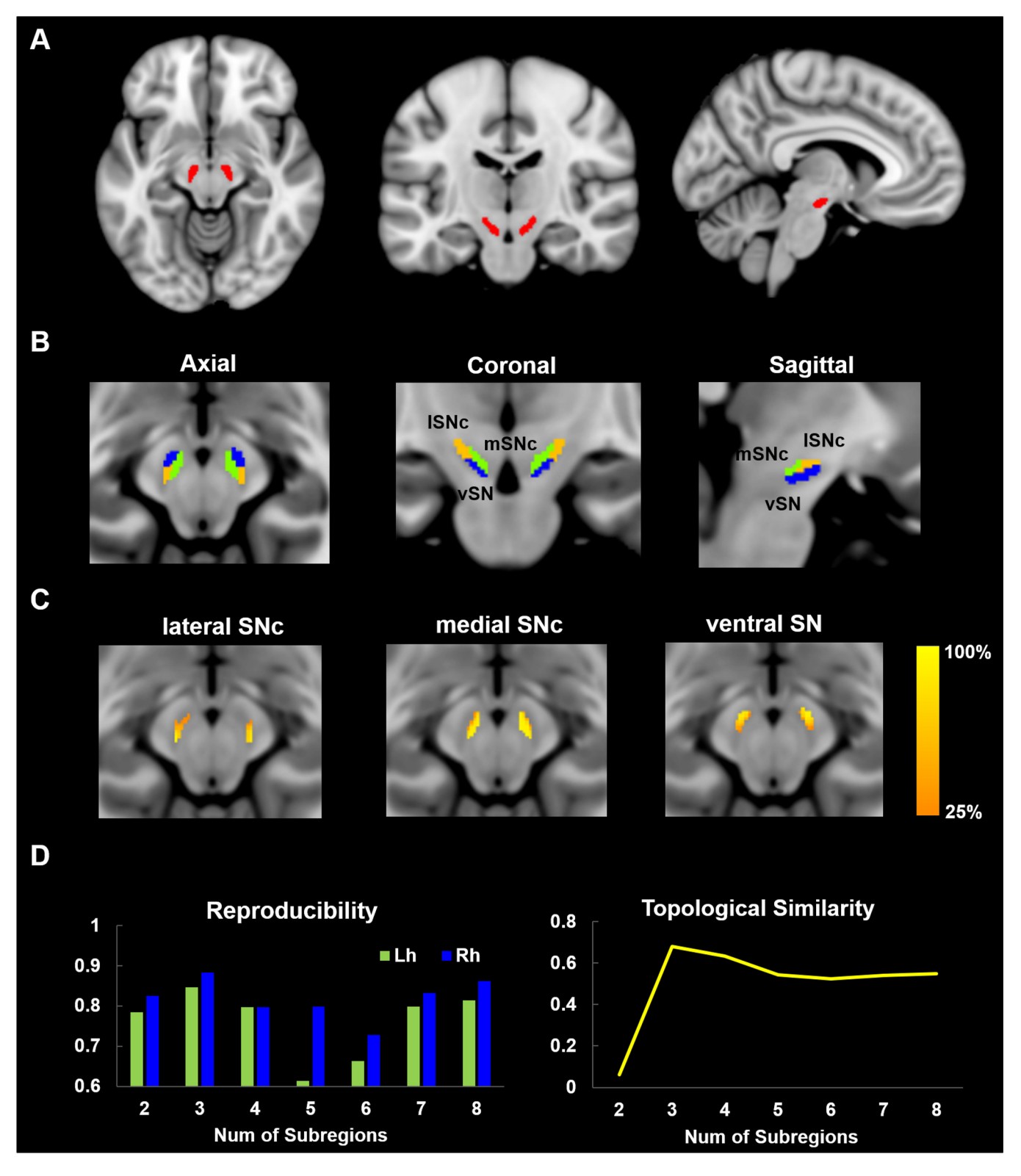

**Figure 2.** Parcellation of Substantia Nigra based on anatomical connectivity profiles. (**A**) Definition of the seed region. Substantia Nigra mask was extracted from a 7T atlas of Basal ganglia based on high-resolution MP2RAGE and FLASH scans (*Keuken and Forstmann, 2015*). (**B**) Parcellation map of SN on 60 healthy young subjects. SN was subdivided into three subregions: a dorsolateral area corresponding to lateral part of SN pars compacta (lSNc), a dorsomedial area corresponding to medial part of SNc (mSNc) and a ventral area (vSN). (**C**) Probabilistic map of each SN subdivision, where

*Figure 2 continued on next page*

*Figure 2 continued*

the intensity at each SN voxel represents the probability of subdivision assignment over the population. A probability of 100% at voxel *i* means that the same assignment is made for every subject. (D) Reproducibility and topological similarity of SN parcellation. The three-cluster parcellation of SN showed both high reproducibility, as assessed by repeated split-half resampling (mean NMI = 0.85 and 0.88, respectively for left and right SN) and high inter-hemispheric topological similarity (mean NMI = 0.68). NMI: normalized mutual information.

DOI: https://doi.org/10.7554/eLife.26653.005

The following figure supplements are available for figure 2:

**Figure supplement 1.** Comparison of the SN parcellation from the current study and the subdivisions of dopamine cells in monkeys.

DOI: https://doi.org/10.7554/eLife.26653.006

**Figure supplement 2.** Replication of the SN parcellation in two independent groups.

DOI: https://doi.org/10.7554/eLife.26653.007

**Figure supplement 3.** Reproducibility and topological similarity in connectivity-based parcellation of bilateral SN measured by Dice coefficient and Cramer's V.

DOI: https://doi.org/10.7554/eLife.26653.008

caudate and anterior part of putamen (associative striatum), and lateral SNc connecting via the posterior limb of the internal capsule with the posterior part of putamen (motor striatum).

A similar organizational pattern was also revealed by analyzing the SN connections to seven canonical resting-sate networks (*Yeo et al., 2011*), with clear dissociations, as well as overlaps, of fiber projections among the three SN subdivisions (*Figure 5—figure supplement 2*). Specifically, medial SNc preferentially connected with the limbic and visual networks; vSN dominantly connected with the frontoparietal and default-mode networks; and lateral SNc mainly connected with the somatomotor and dorsal attention networks.

Finally, connectivity fingerprints of the three SN subdivisions were generated by mapping the connectivity profiles to a fine-grained whole brain anatomical connectivity atlas (*Fan et al., 2016*). As shown in *Figure 5*, the three subdivisions of SN showed distinct connectivity profiles in frontal, parietal, temporal and subcortical areas. Specifically, most prefrontal areas showed the strongest connections with ventral SN (*Figure 5—figure supplement 1A*), except for several subregions in superior and middle frontal gyrus, for instance lateral and medial area 6 (i.e. areas SFG_c7_4/5 and MFG_c7_6), more strongly connecting with lateral SNc. The somatic motor and sensory cortex preferentially connected with lateral SNc (*Figure 5—figure supplement 1B*). Meanwhile, most limbic regions were targeted by fiber tracts derived from medial SNc (*Figure 5—figure supplement 1C*).

In summary, all the connectivity profiles of the SN subdivisions are consistent with a limbic (medial SNc), motor (lateral SNc), and cognitive (ventral SN) functional organization.

**Table 1.** Regions of interest used in this study and their BOLD activity during the gambling task.

| Brain regions | X | Y | Z | Brain Activity (T-score) | | |
|---|---|---|---|---|---|---|
| | | | | Reward | Punishment | RPE |
| SN subdivisions | | | | | | |
| vSN | ±9 | −13 | −12 | 11.12 ** | 8.63 ** | 1.85 |
| medial SNc | ±8 | −16 | −12 | 17.45 ** | 12.48 ** | 3.96 ** |
| lateral SNc | ±12 | −17 | -9 | 9.72 ** | 8.87 ** | 0.93 |
| Ventral Striatum | ±12 | 15 | -6 | 8.91 ** | - 9.33 ** | 16.25 ** |
| vmPFC | ±6 | 45 | -9 | - 26.72 ** | - 32.37 ** | 8.75 ** |
| Anterior insula | ±32 | 22 | -6 | 35.54 ** | 36.62 ** | 1.07 |
| dACC | ±4 | 40 | 24 | 8.43 ** | 7.64 ** | 1.37 |

Notes: **p-value<0.01; *p-value<0.05 with FDR correction

SN: substantia nigra; vSN: ventral subregion of SN; SNc: SN pars compacta; vmPFC: ventral medial prefrontal cortex; dACC: dorsal anterior cingulate cortex; RPE: reward prediction error

DOI: https://doi.org/10.7554/eLife.26653.009

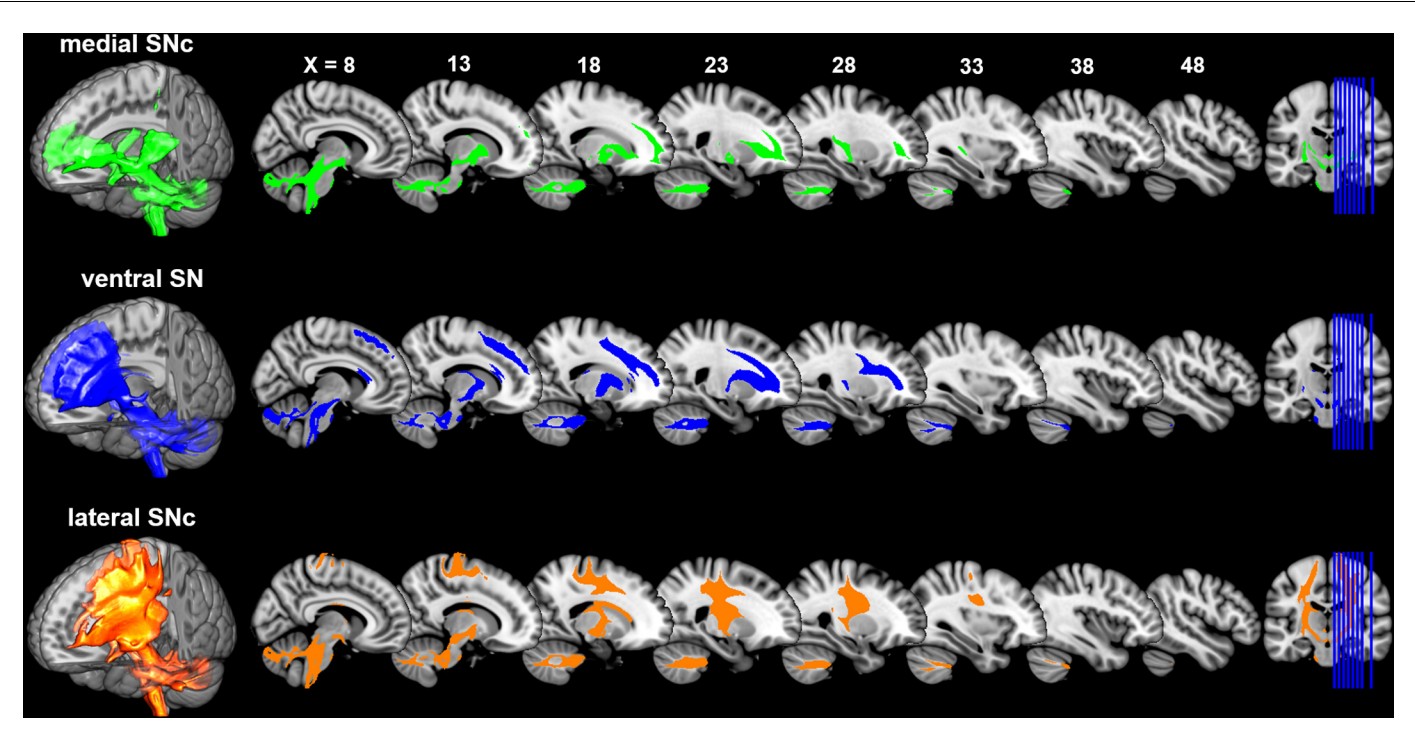

**Figure 3.** Connectivity patterns of the subdivisions of substantia nigra. Probabilistic fiber tractrography was performed for each SN subdivision to map its whole-brain connectivity patterns. The population tract maps are shown with a threshold of connectivity probability at 0.05 and rendered using MRIcron on the ICBM152 brain template.

DOI: https://doi.org/10.7554/eLife.26653.010

The following figure supplements are available for figure 3:

**Figure supplement 1.** Distinct and overlapping tractography maps among the subdivisions of substantia nigra.

DOI: https://doi.org/10.7554/eLife.26653.011

**Figure supplement 2.** Connectivity profiles revealed by deterministic fiber tractography using DSI-studio (http://dsi-studio.labsolver.org/) based on the group averaged template of diffusion data from HCP-500 subjects.

DOI: https://doi.org/10.7554/eLife.26653.012

## Brain activity during gambling task

In order to ascertain different functional roles of SN subdivisions, we explored their BOLD response to win and loss outcomes during the fMRI gambling task. Brain activation maps of value-coding (i.e. contrast of the difference in response to reward versus punishment) and salience-coding (i.e. contrast of the mean response to reward and punishment versus neutral) were assessed by whole-brain analysis using one-sample t-tests and corrected for multiple comparisons using the threshold-free cluster enhancement method (*Smith and Nichols, 2009*). Significant value-coding was detected in ventral striatum and vmPFC, while salience signals were found in anterior insula, dorsal anterior cingulate cortex (dACC) and dorsal striatum (*Figure 6—figure supplement 1*). Whole-brain contrast analysis of value- and salience-related BOLD responses provided further evidence for the dissociation of reward value and motivational salience (*Figure 6—figure supplement 2*).

Next, ROI-based analysis was performed on the three SN subdivisions and four presumed target regions: ventral striatum, vmPFC, anterior insula and dACC. Two-way repeated measures ANOVA revealed a significant interaction effect between SN subdivisions and BOLD response to reward or punishment (F = 6.6, p=0.0014). As shown in *Figure 6*, all three SN subregions were activated by both reward and punishment, but only medial SNc showed significantly greater neural activity to monetary gains than losses (T = 3.96, p<0.0001). Moreover, among the subdivisions of SN, medial SNc showed significantly higher BOLD response to positive value (gains) compared to the other two subdivisions (T = 3.52, p=0.0005 for vSN; T = 2.98, p=0.003 for lateral SNc), as shown in *Figure 6—figure supplement 3*.

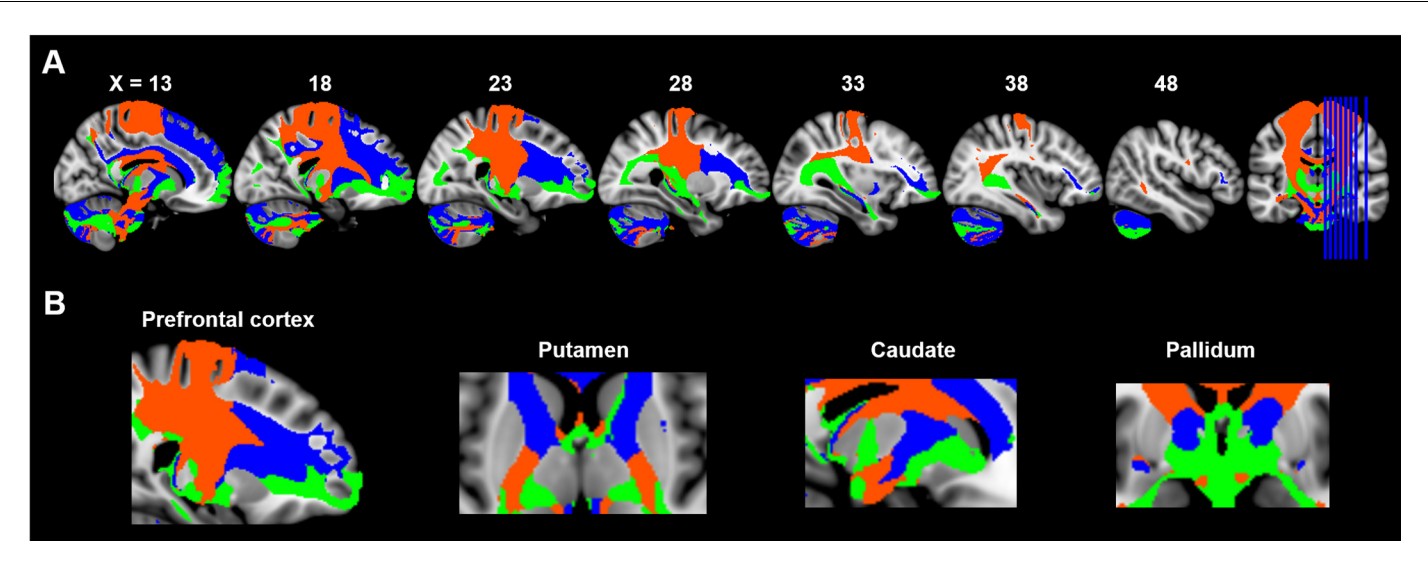

**Figure 4.** Maximum probability tractograms of the subdivisions of substantia nigra. A limbic-cognitive-motor organizational topography of SN projections is shown in multi-slice views (**A**) with a focus on prefrontal cortex, striatum and pallidum (**B**). Maximum probability tractograms were generated by assigning each voxel to the corresponding SN subdivision with which it showed the greatest connections. SN subdivisions: vSN (blue), mSNc (green) and lSNc (orange).

DOI: https://doi.org/10.7554/eLife.26653.013

The following figure supplements are available for figure 4:

**Figure supplement 1.** Maximum Probability Map of tractograms from SN subdivisions under different thresholds of tractography probability.

DOI: https://doi.org/10.7554/eLife.26653.014

**Figure supplement 2.** Distinct and overlapping projections of SN subdivisions in basal ganglia.

DOI: https://doi.org/10.7554/eLife.26653.015

A significant interaction effect was also detected between the four target regions and their BOLD response (F = 30.26, p-value=3e-19). Specifically, ventral striatum showed a unique bi-directional pattern, i.e. strongly activated during reward (T = 8.91, p<0.0001) and deactivated during punishment (T = −9.33, p<0.0001). Meanwhile, as a core region in the default mode network (DMN; [*Buckner et al., 2008*; *Raichle, 2015*]), vmPFC was deactivated in both conditions (T = −26.72 and −32.37 with p<0.0001 for reward and punishment, respectively). Both ventral striatum and vmPFC showed significantly greater response to rewarding than aversive outcomes (T = 16.25, p<0.0001 for ventral striatum, T = 8.75, p<0.00001 for vmPFC), although in the case of vmPFC this consisted of less deactivation (*Figure 6*). Of all regions examined, only VS showed the classic reward prediction error pattern (activation for gains, inhibition for losses). On the contrary, as the core areas of the salience network (*Seeley et al., 2007*), dACC and anterior insula were activated by both types of trials, with no difference in response to reward and punishment (T = 1.37, p=0.17 for dACC, T = 1.07, p=0.28 for anterior insula). These results support the theory that there are at least two separate brain dopamine-related systems involved during gambling outcomes, with one encoding reward value (i.e. different response to reward and punishment) and the other encoding motivational salience (i.e. reacting similarly to rewarding and aversive outcomes).

## Correlation analysis between brain activity and impulsivity measures

We next sought to determine if brain activity in the value and salience coding systems was related to two impulsive traits: decisional and motor impulsivity. Based on the above brain connectivity (*Figures 3–5*) and activity analyses (*Figure 6*), we take the value-coding system to consist of mesolimbic pathways projecting between medial SNc and ventral striatum and vmPFC, and the salience-coding system of mesocortical pathways connecting vSN with dACC and anterior insula. We correlated BOLD activity of these brain areas with behavioral measures of decisional and motor impulsivity. We

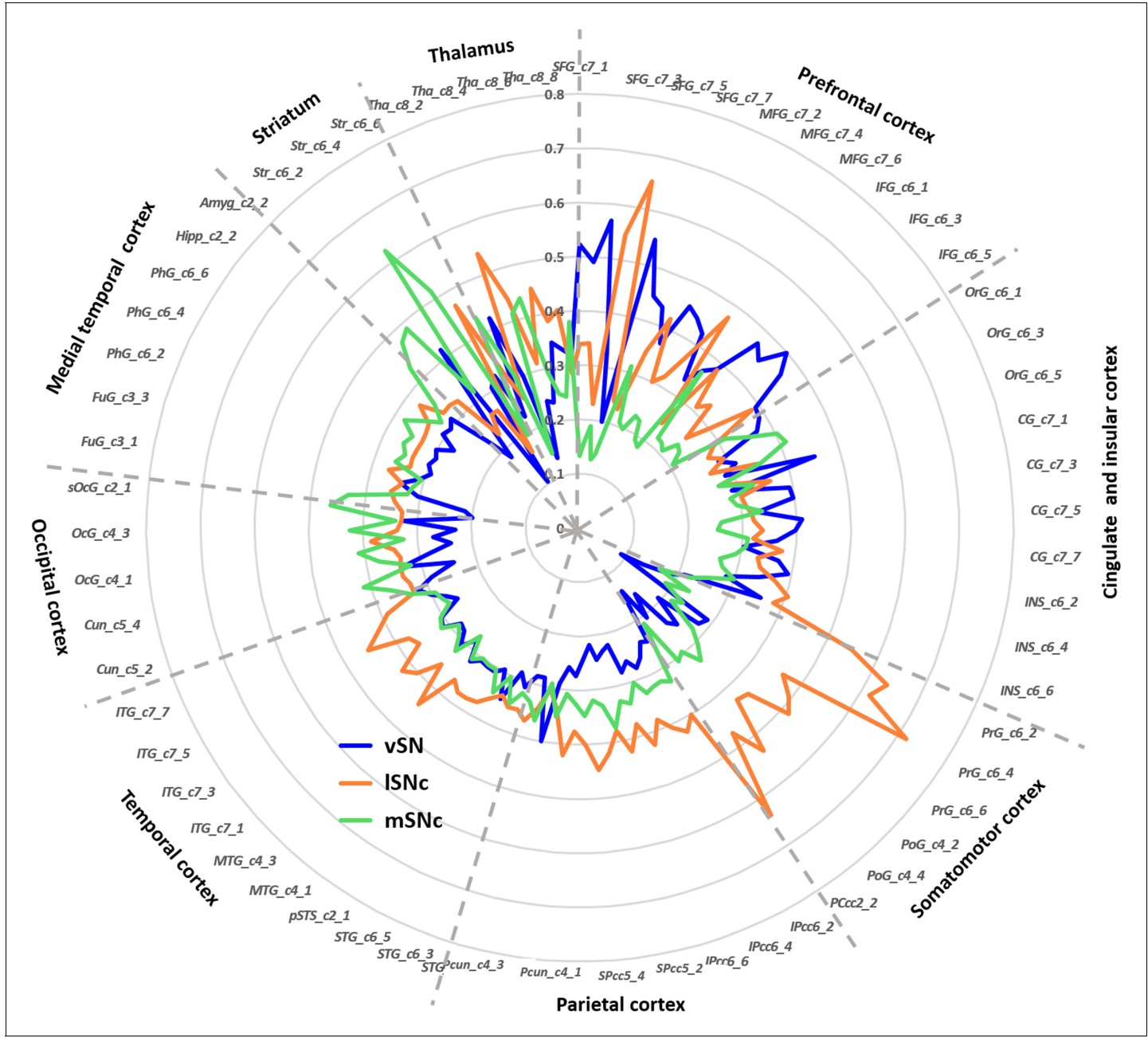

**Figure 5.** Connectivity fingerprints of the subdivisions of Substantia Nigra. The connectivity fingerprints of SN subdivisions were calculated based on a whole-brain atlas (*Fan et al., 2016*). The relative connectivity strength between each target (i.e. parcels in the brain atlas) and each SN subdivision is plotted. An organizational topography of SN projections emerges with vSN mostly connected to prefrontal cortex, lateral SNc to sensorimotor cortex, and medial SNc to limbic regions. The naming convention is based on *Fan et al. (2016)*. The atlas is available at http://atlas.brainnetome.org/.
DOI: https://doi.org/10.7554/eLife.26653.016

The following figure supplements are available for figure 5:

**Figure supplement 1.** Connectivity profiles of SN subdivisions in prefrontal, motor and limbic systems.
DOI: https://doi.org/10.7554/eLife.26653.017
**Figure supplement 2.** Connectivity profiles of SN subdivisions to seven resting-state functional networks.
DOI: https://doi.org/10.7554/eLife.26653.018
**Figure supplement 3.** Target regions of the connectivity fingerprints.
DOI: https://doi.org/10.7554/eLife.26653.019

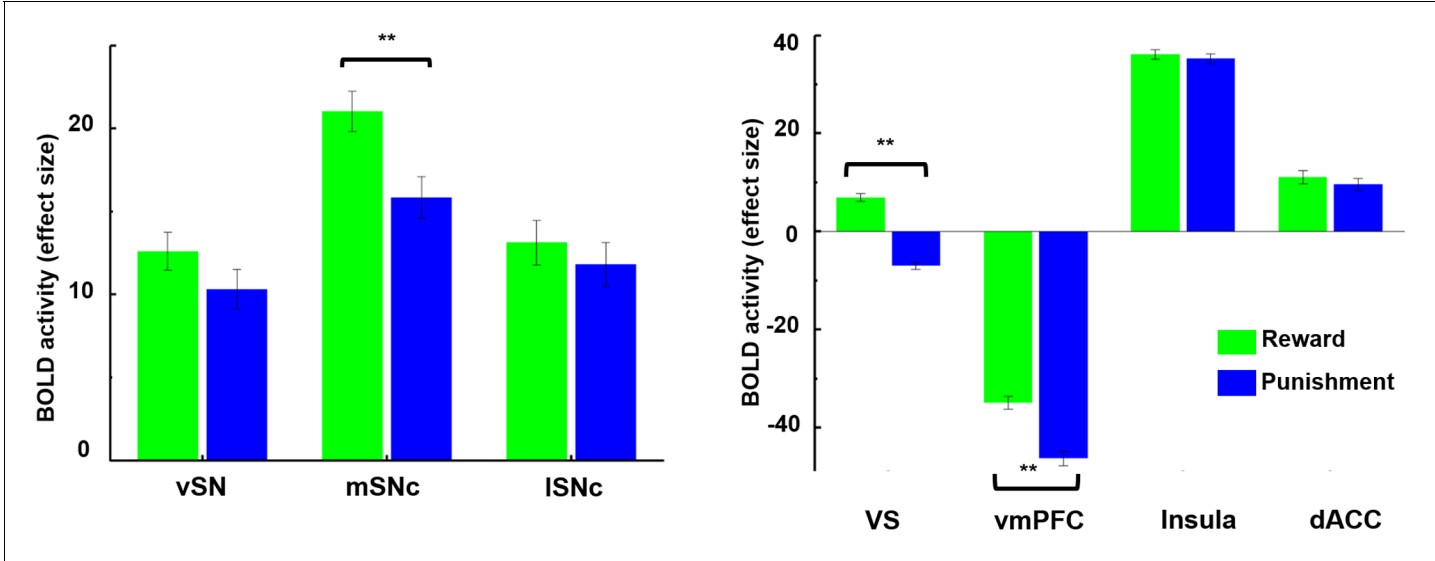

**Figure 6.** Brain activity in response to rewarding and aversive outcomes in the fMRI gambling task. Among SN subdivisions, only medial SNc showed a significant difference in response to reward and punishment (p<0.001). The ventral striatum (VS) and ventromedial prefrontal cortex (vmPFC) also responded differently to reward and punishment, with greater BOLD activity to rewarding than aversive stimuli (p<0.001). Meanwhile, anterior insula and dorsal anterior cingulate cortex (dACC) showed no difference in response to reward and punishment.

DOI: https://doi.org/10.7554/eLife.26653.020

The following figure supplements are available for figure 6:

**Figure supplement 1.** Whole-brain analysis of BOLD response to value-coding and salience-coding.

DOI: https://doi.org/10.7554/eLife.26653.021

**Figure Supplement 2.** Contrast between value- and salience-related BOLD responses.

DOI: https://doi.org/10.7554/eLife.26653.022

**Figure supplement 3.** Brain activity in response to value and salience.

DOI: https://doi.org/10.7554/eLife.26653.023

reasoned that decisional impulsivity would implicate the value-coding dopamine system, while motor impulsivity would implicate salience or motor system projections.

To characterize the relationship between BOLD activity and impulsivity measures, behavioral PLS analysis (*McIntosh and Lobaugh, 2004*) was performed. BOLD effect sizes from all seven regions that were detected as significantly activated during either value- or salience-coding (i.e. mSNc, VS and vmPFC for value-coding; all target regions except VS for salience-coding) were imported as the brain data. The two behavioral measures of impulsivity were imported as the behavior data. Behavioral PLS has the potential to identify commonalities and differences across conditions in brain-behavior relations. Fifteen subjects were excluded before the analysis to ensure that all subjects had complete records of both behavioral measures within two standard deviations of the mean.

Two components were identified (*Figure 7*, proportion of covariance = 61% and 39%, permuted p-value=0.012 and 0.11, respectively), with one significantly correlated with inhibitory control scores (r = 0.1896, CI= [0.1575, 0.2781]) while the other was significantly correlated with Delay-Discounting measures (r = −0.1535, CI= [−0.2499,–0.1277]). The reliability of both brain and behavior loadings was assessed by estimating their confidence intervals using bootstrap resampling. The first component (inhibitory control) comprised the value signal from vmPFC (z-score of weights = 2.1482) and salience signal from insula (z-score = 2.5006) and dACC (z = 2.7626). The second component (delay-discounting) comprised the value signal from mSNc (z-score = 2.0518) and VS (z-score = 2.5937). Associations between brain response and behavioral scores were found for both components (r = 0.1706 and 0.1551, respectively). The latent variables identified by PLS (including both brain and behavioral scores) are intrinsically orthogonal across different components. The resulting brain networks and behavioral measures are consequently independent if they belong to different components.

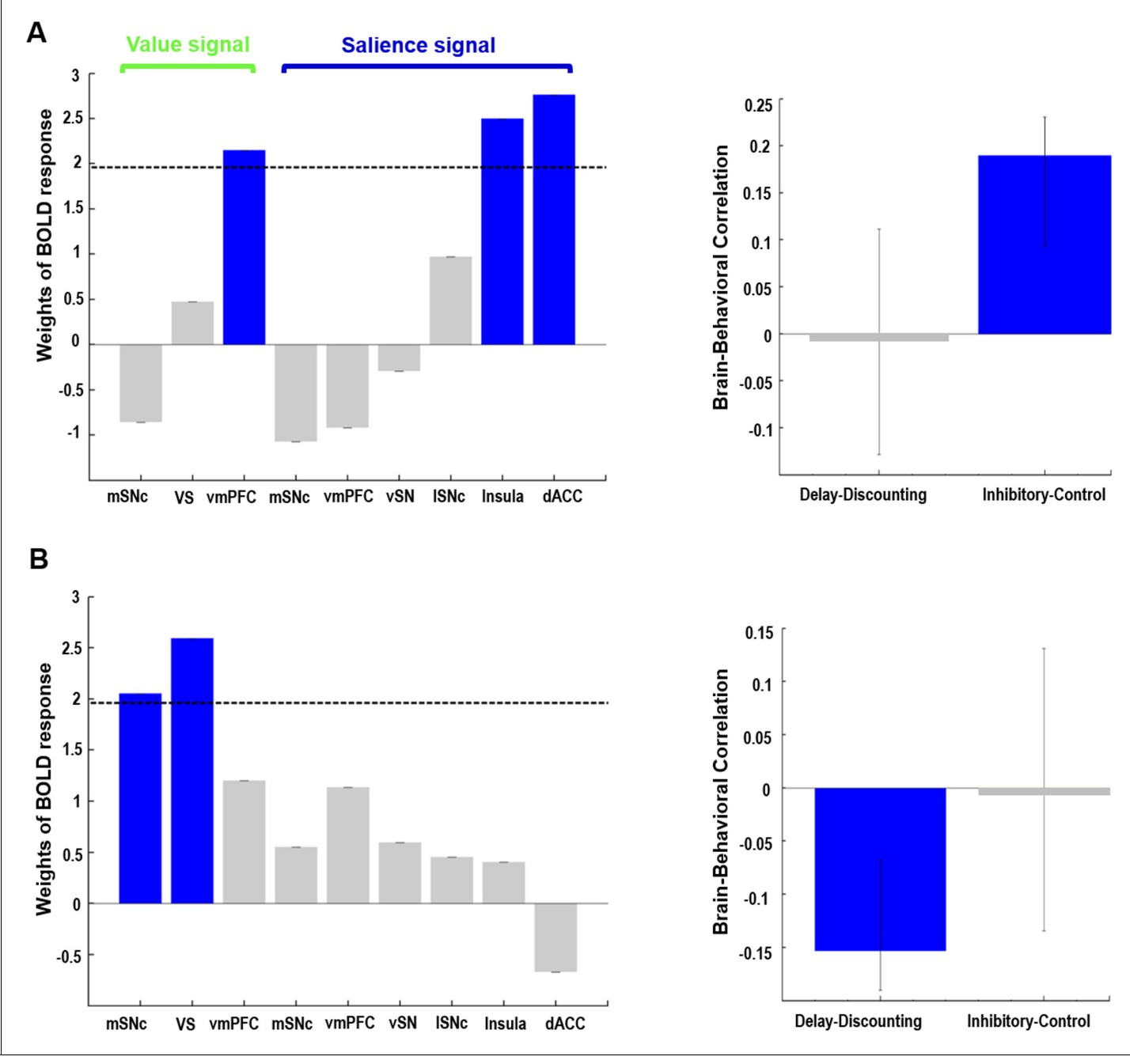

**Figure 7.** Behavioral PLS analysis on value- and salience-coding BOLD response and two measures of impulsivity. Two orthogonal components were identified. The first component (**A**) captured the relationship between value signal from vmPFC, salience signals from anterior insula and dACC, and inhibitory control scores on the Flanker task. The second component (**B**) captured the relationship between value signals from mSNc and VS, and the AUC measure of delay discounting. Value and salience BOLD signals were derived from the gambling task. Note: for delay-discounting lower AUC indicates greater impulsivity. (AUC: area under the curve). In the leftmost panel, blue color of the bars indicates a reliable contribution (z > 1.96) as determined by the bootstrap procedure. Error bars in the middle panel represent 95% confidence intervals derived from bootstrap resampling. See text for abbreviations.

DOI: https://doi.org/10.7554/eLife.26653.024

The following figure supplement is available for figure 7:

**Figure supplement 1.** Correlation analysis between value and salience BOLD response and two measures of impulsivity.

DOI: https://doi.org/10.7554/eLife.26653.025

Individual correlation analyses between BOLD responses and impulsivity measures confirmed the PLS results (*Figure 7—figure supplement 1*). We found significant correlations between AUC of Delay Discounting and value-related BOLD response in medial SNc (r = −0.1304, p=0.0053) and ventral striatum (r = −0.1191, p=0.01), and correlations between inhibitory control scores and salience-related BOLD response in dACC (r = 0.1112, p=0.0161) and anterior insula (r = 0.1546, p=0.0008). Only the associations of value-coding response in mSNc with Delay-Discounting and salience-coding response in insula with inhibitory control remain significant after Holm–Bonferroni correction with p-value=0.05. Moreover, the correlation of value-coding response in VS was significantly higher with Delay-Discounting measures than inhibitory control scores (p=0.02), while the association of salience-coding response in anterior insula was significantly stronger with inhibitory control than Delay-Discounting (p=0.008).

## Discussion

### Subdivisions of SN

We used a connectivity-based parcellation scheme to subdivide human SN based on its anatomical connectivity profile with the rest of the brain. A tripartite pattern of SN was revealed, consisting of a medial (mSNc) tier, a lateral (lSNc) tier and a ventral (vSN) tier. A similar anatomical and connectional differentiation of SN has been widely described in monkeys. Indeed, many studies report that midbrain dopamine neurons can be divided into two or three tiers (*François et al., 1999*; *Haber and Knutson, 2010*; *Lynd-Balta and Haber, 1994*), with a dorsal calbindin-positive tier that extends medially to the VTA, and a ventral calbindin-negative tier whose dendrites extend ventrally into the pars reticulata of the substantia nigra. This ventral tier can be further subdivided into a more medio-dorsal densocellular group and a ventro-lateral group of columnar cells (*Haber, 2014*). Tracer studies in monkeys have been used to map the striatal afferent and efferent projections of these SN subdivisions (*Haber et al., 2000*; *Haber and Knutson, 2010*; *Lynd-Balta and Haber, 1994*). The dorsal tier mainly connects with ventromedial striatum, while the ventral tiers project to central and dorsolateral striatum. Coinciding with monkey anatomy, we also found a tripartite division of SN with similar anatomical and connectivity profiles (*Figure 2B* and *Figure 2—figure supplement 1*). Specifically, our mSNc corresponds to the monkey pars dorsalis and connects with ventral striatum; lSNc corresponds to the ventrolateral columnar part of SNc and connects to the motor regions of dorsal striatum. Finally, our vSN corresponds to the ventral densocellular portion that projects to the middle, associative, part of the striatum (*Haber, 2014*). Furthermore, the cortical projections of SN subdivisions we identified using diffusion tractography also fit with this limbic (mSNc), associative (vSN), and somatomotor (lSNc) organization (*Figures 3–5*). This specific association of lSNc with sensorimotor cortex explains its crucial role in the motor symptoms of Parkinson disease, in which ventrolateral SNc is preferentially targeted by neurodegeneration (*Gibb and Lees, 1991*). It is worth mentioning that this organizational pattern of SN projections (*Figure 4*) was detected by using a winner-takes-all approach, which emphasizes the distinct connections among subdivisions. A more lenient threshold for the identification of connections reveals large overlaps of anatomical connections among SN subdivisions (*Figure 3—figure supplement 1*). The three SN subdivisions have different but overlapping connectional profiles (*Figure 5*). For example, while SN-vmPFC connections were mostly derived from medial SNc, the other two subdivisions also made considerable contributions. This is consistent with the view that vmPFC is a connectional hub that integrates broad domains of information to support the valuation process during decision-making (*Benoit et al., 2014*; *Roy et al., 2012*).

The inverted dorsal/ventral topography of SN-striatum connections (*Haber, 2014*) found here has previously been described in human brain (*Chowdhury et al., 2013*). There, the authors used diffusion tractography in 30 individuals to parcellate SN based on anatomical connections with two targets in striatum. The dorsal SN mainly connected to ventral striatum, while the ventral SN preferentially connected to dorsal striatum. In contrast to this study, we used the whole-brain connectivity profiles to identify the subareas within SN instead of using predefined regions of interest restricted to striatal regions.

Recently, whole-brain tractography was also performed on the HCP dataset to identify the major brainstem white matter tracts (*Meola et al., 2016*). Two distinct fiber tracts were found projecting

through SN: the frontopontine tract (FPT) connecting prefrontal cortex and anterior SN and running through the anterior limb of the internal capsule, and the corticospinal tract (CST) connecting motor cortex and posterior SN and passing through the posterior limb of the internal capsule. This accords with our fiber-tracking results, with vSN mainly connecting to the prefrontal cortex through the anterior limb of the internal capsule adjacent to the anterior dorsal striatum (including the body of caudate and anterior part of putamen), and lSNc preferentially connecting to the sensorimotor cortex via the posterior limb of the internal capsule and adjacent posterior dorsal striatum (including the tail of caudate and posterior putamen). We identified an additional fiber tract, i.e. a mesolimbic pathway connecting medial SNc with ventral striatum, vmPFC and OFC. On the other hand, some detected connections of SN were unexpected. For instance, the midbrain projections to visual cortex are sparse in rodents (*Watabe-Uchida et al., 2012*), but we detected relatively strong anatomical connections between mSNc and visual areas using both probabilistic and deterministic tractography (*Figure 3—figure supplements 1* and *2*). This might be caused by an intersection of mSNc outflow with the temporo-parieto-occipito-pontine tract in the posterior limb of the internal capsule (*Meola et al., 2016*).

The somatomotor to associative to limbic (from lateral to medial) organization of SN accords with the cortical arrangement of information flow proposed by Mesulam (*Mesulam, 1998*), in which unimodal areas project to heteromodal associative, and then to prelimbic and limbic regions. A recent study proposed a similar gradient of cortical information processing based on resting state fMRI data from the HCP (*Margulies et al., 2016*). Our results suggest that the somatomotor to associative to limbic principle of cortical organization appears to be reflected in the SN.

## Value and salience coding in SN projections

We found a dissociation between coding of value and salience within SN subdivisions and their projections. While all three subdivisions responded to both winning and losing money during the gambling task (*Figure 6*), only mSNc encoded a classical value signal, showing significantly greater BOLD response to wins than losses (*Figure 6*). Medial SNc preferentially connects to limbic areas including ventral striatum, ventral pallidum, hippocampus, amygdala and OFC/vmPFC (*Figure 4*). These brain regions have been reported to support value-based reinforcement learning (*Garrison et al., 2013*; *Glimcher, 2011*), and goal-directed behaviors (*Goto and Grace, 2005*), and have been implicated in drug addiction (*Nutt et al., 2015*). By contrast, vSN encoded salience, showing a similar BOLD response to rewarding and aversive events (*Figure 6*). Ventral SN mainly connects to the prefrontal cortex and salience network, including lateral frontal cortex, dorsomedial prefrontal cortex, dACC and anterior insula (*Figure 4*). These brain areas are associated with attention, orientation and cognitive control (*Menon and Uddin, 2010*; *Seeley et al., 2007*; *Uddin, 2015*). Finally, the lSNc subdivision also appeared to encode salience, responding equally to monetary gains and losses. In contrast to the mesocortical pathway derived from vSN, the predominant projections of lSNc were with the motor cortex, premotor cortex, supplementary motor area, and posteriorly into the parietal cortex (*Figure 4*). The value/salience dissociation of lateral and medial SNc corresponds to the findings from recordings in Macaque midbrain dopamine neurons (*Bromberg-Martin et al., 2010*; *Matsumoto and Hikosaka, 2009*). A somewhat similar functional dissociation in SN was reported in a fMRI study with a Pavlovian learning paradigm (*Pauli et al., 2015*), with the lateral SN encoding a prediction signal for aversive events, and the medial SN encoding a reward prediction error signal for appetitive learning.

## SN and impulsivity

Impulsivity has been reported to contribute to a wide range of psychopathology including bipolar disorder (*Swann et al., 2009*), ADHD (*Winstanley et al., 2006*), alcohol and substance dependence (*Ersche et al., 2010*), pathological gambling (*Leeman and Potenza, 2012*) and addictive behaviors in Parkinson's disease (*Averbeck et al., 2014*; *Dagher and Robbins, 2009*). A current account of impulsivity assigns a key role to midbrain dopamine neurons, which modulate choice behaviors through the direct and indirect corticostriatal pathways (*Buckholtz et al., 2010*; *Dalley and Roiser, 2012*). Specifically, phasic bursts of dopamine firing enhance impulsive and risk-taking behaviors through D1 receptors within the direct pathway, while pauses in dopamine firing activate inhibitory

control through D2 receptors within the indirect pathway (*Collins and Frank, 2014*; *Cox et al., 2015*). Impulsivity may also be a reflection of top-down cortical and striatal control of SN activity.

The multi-dimensional view of impulsivity proposes at least two major components (*Meda et al., 2009*), namely impulsive action and impulsive choice. Here, we included two different impulsivity measures, the Delay Discounting task (impulsive choice) and the Flanker inhibitory control task (impulsive action), and explored the neural basis of these two constructs.

Our results suggest that two different dopamine systems modulate these two components of impulsivity in parallel. Specifically, decisional impulsivity, measured by the Delay Discounting task, was associated with the value-coding system comprising mSNc and VS (*Figure 7*). Stronger value-coding signals in these areas were associated with more impulsive choices during delay discounting, meaning higher preference for immediate and smaller rewards. Meanwhile, motor impulsivity measured by the Flanker inhibitory control task was associated with the salience-coding system consisting of dACC and anterior insula (*Figure 7*). Stronger BOLD signals in the salience network predicted better attentional inhibitory control. This finding is consistent with the theory that anterior insula plays an important role in inhibitory control by increasing the saliency of stimuli, especially for unexpected events (*Cai et al., 2014*; *Ghahremani et al., 2015*).

## Limitations

We included a large population of healthy young subjects acquired from the public HCP dataset. Multimodal data included structural, diffusion-weighted and functional MRI, as well as behavioral impulsivity measures. There were a few missing imaging or behavioral data and some datasets failed during additional preprocessing. The final dataset included 485 subjects for the gambling-task fMRI data, 430 subjects for the diffusion data, and 488 subjects for the behavioral measures. In the end, we had over 400 overlapping subjects who had all three modalities.

The SN is a small nucleus located in the brainstem, where MRI data usually suffer from distortions and signal losses. Partial volume effect might have impacted the imaging data, especially for fMRI. However, in the HCP data, these problems have been mitigated by advanced high-resolution imaging sequences and preprocessing (*Glasser et al., 2013*; *Sotiropoulos et al., 2013*). Still, one potential limitation of the current study is inferring midbrain dopaminergic projections from diffusion MRI. Diffusion tractography has several known limitations, including the inability to perfectly resolve crossing fibers, a relatively high susceptibility to false positives and negatives and a tendency to terminate in gyral crowns as opposed to sulci, resulting in diminished anatomical accuracy (*Jbabdi et al., 2015*; *Jones et al., 2013*; *Thomas et al., 2014*). A greater concern, however, is the possibility of systematic biases in probabilistic tractography that may give an incorrect impression of whole-brain SN connectivity patterns. For example, it is accepted that connections are less likely to be detected if they travel a long distance, exhibit marked curvature or branching, travel close to cerebrospinal fluid, or pass through more complex white matter regions (*Jbabdi et al., 2015*). Most of our results are unlikely to be due to such biases. First, although the SN parcellation and projection maps were based on diffusion tractography, they also reveal a functional dissociation. That is, projection maps of SN subdivisions reflect a limbic, associative and somatomotor organization, rather than a purely geometric pattern. Moreover, our parcellation of SN accords closely with tract tracing studies in macaque (*Haber, 2014*). The inverted dorsal-ventral topology of SN-striatum connections (*Figure 4*) is difficult to account for based on a distance or curvature bias; however, the medial-lateral gradient of SN projections with basal ganglia described here could reflect such a limitation. We found that medial SNc mainly connects with the medial part of nucleus accumbens and globus pallidus, while lateral SNc predominantly connects with lateral dorsal striatum, including tail of caudate and posterior putamen. This medial-lateral arrangement could result from a bias of tractography. While the SN-striatum arrangement is in accordance with known connectivity in macaque (*Haber et al., 2000*; *Haber and Knutson, 2010*; *Lynd-Balta and Haber, 1994*) and other species (*Düzel et al., 2009*), it is possible that SN-pallidal connections described here are artifactual, as they do not accord with previously documented anatomical connections of basal ganglia. The pattern of SN-striatum connections is also consistent with the presumed functional roles of medial (limbic) and lateral (motor) structures in both SN and striatum. Regarding SN-cortical connections, the predominant topology found here is not medial-lateral. For example, the lateral SNc connects mostly medially in primary and supplementary motor areas (*Figures 3–4*). Indeed, the topological arrangement of SN-cortical connections exhibits a rotation from medio-lateral in SN to ventro-dorsal (and rostro-caudal) in cortex

(*Figure 3*), as described previously for tractography of the ventral prefrontal cortex (*Jbabdi et al., 2013*). Another limitation is that connectivity measured by diffusion tractography cannot resolve the direction of a connection, which makes it impossible to distinguish efferent dopamine projections from top-down fronto-nigral or striato-nigral projections.

Thus, although our results are consistent with those of Matsumoto and Hikosaka in monkeys (*Matsumoto and Hikosaka, 2009*), in which there is a medial to lateral gradient for reward/salience coding, we cannot attribute either connectivity or fMRI activation to dopamine neurons per se. The population of neurons in VTA and SN is heterogeneous and includes GABAergic and glutamatergic projection neurons and interneurons (*Henny et al., 2012*; *Morales and Margolis, 2017*). Thus, despite a reward/salience dissociation found in SN activation and projections, our results do not contradict the theory that dopamine neurons in SN/VTA are predominantly excited by reward and reward prediction error (*Cohen et al., 2012*; *Fiorillo, 2013*). It is also notable that the fMRI results demonstrated a salience response (i.e. to both wins and losses) throughout the SN, consistent with recordings in monkeys, and that our connectivity findings support the theory that predominantly reward versus predominantly salience coding SN neurons belong to different brain networks (*Bromberg-Martin et al., 2010*). Note, however, that our inability to assign activation or connectivity to dopamine neurons (rather than SN neurons generally) means that our results do not contradict the alternative interpretation that no dopamine neurons encode actual aversive value (*Fiorillo, 2013*; *Schultz et al., 2017*). These authors suggest that dopamine neurons have an initial positive short-latency response to all stimuli that depends on their intensity, but that the subsequent response always encodes true value (activation for rewards, suppression for punishments).

Finally, we describe associations between behavioral measures of impulsivity and BOLD activation during the gambling task. While these findings indicate a relationship between intrinsic organization of SN-related brain networks and impulsivity, the correlation values are small (on the order of 0.1). Therefore, most of the variability in behavior is accounted by other factors.

## Conclusions

We subdivided the human SN into three subpopulations according to anatomical connectivity profiles, with a dorsal-ventral and lateral-medial arrangement. Our three-way partition of SN reveals multiple dopaminergic systems in human SN, showing a limbic, cognitive and motor arrangement, and encoding value and salience signals separately through distinct networks. Corresponding to this connectional arrangement, we also found dissociable functional response during the gambling task and correlations with impulsivity measures. Specifically, the mSNc-VS system was involved in value-coding and associated with impulsive choice, while the vSN-dACC-insula system was involved in salience-coding and associated with response inhibition. Building on the traditional reward prediction error model of dopamine signaling (*Schultz, 1998*), our study provides evidence for the connectional and functional disassociations of SN neurons in humans, which encode motivational value and salience, possibly through different dopaminergic pathways. We also extended the current view on the role of dopamine in impulsivity by uncovering different neural substrates for decisional and motor impulsivity.

## Materials and methods

### Subjects and data acquisition

Data from 485 healthy individuals (age: 29.1 ± 3.5 years, 202 females) were obtained from the 500-subject release of the Human Connectome Project (HCP RRID:SCR_008749) database from March 2015. Participants with body mass index (BMI) lower than 18 were considered as underweight and excluded from this study. Multimodal data used here include structural MRI, diffusion-weighted MRI, functional MRI during a gambling task and behavioral measures of impulsivity. The scanning procedures are described in detail in *Van Essen et al. (2013)* and available online (https://www.human-connectome.org/documentation/S500/HCP_S500_Release_Reference_Manual.pdf).

### Diffusion MRI

Diffusion data were collected with 1.25 mm isotropic spatial resolution and three diffusion weightings using HCP dMRI protocol (*Sotiropoulos et al., 2013*). The data were downloaded in a minimally

pre-processed form using the HCP Diffusion pipeline (*Glasser et al., 2013*) including: normalization of $b0$ image intensity across runs; correction for EPI susceptibility and eddy-current-induced distortions, gradient-nonlinearities and subject motion. Next, the probability distributions of fiber orientation were estimated by using FSL's (RRID:SCR_002823) multi-shell spherical deconvolution toolbox (bedpostx), where each voxel contains at most three fiber directions and the diffusion coefficients were modelled using a Gamma distribution (*Jbabdi et al., 2012*). A T1-weighted image downsampled to the resolution of the diffusion data was employed for the nonlinear registration of the SN seed from MNI standard space to native structural volume space using FNIRT from the FSL package. In total, 430 subjects' data were pre-processed and passed quality control. The parcellation of SN was carried out on 120 randomly selected subjects in order to limit computation time and data storage. To test the robustness of parcellation, we randomly divided the 120-subject dataset in half and applied the parcellation procedure (as explained below) independently in each group. Specifically, the first 60 subjects were used as the test group to reveal the underlying organizational pattern of SN. A second group of 60 subjects was then used as the replication group to test the stability of our parcellation maps. All diffusion data (N = 430) were then used during the tractography analysis to map the connectivity profiles of each of the SN subdivisions identified by parcellation.

## Functional MRI with gambling task

During the gambling task (*Barch et al., 2013*), participants were asked to guess the number on a mystery card. The card numbers ranged from one to nine and participants were asked to guess whether the mystery card number was above or below five by pressing one of two buttons. The outcome of each trial were either winning $1.0 or losing $0.50. Neutral trials with no gain or loss were also included. Participants received their net winnings after completing the task. More details about the task design and data acquisition can be found in (*Barch et al., 2013*). A general linear model (GLM) implemented in FSL's FILM (*Woolrich et al., 2001*) was used to estimate the neural activity during feedback by convolving rewarding and losing trial outcome times with a double gamma 'canonical' hemodynamic response function. The resulting parameter estimates (cope) of contrast images from the two acquisitions using different phase encoding directions (i.e. LR and RL coding) were combined to generate individual BOLD activity during outcomes (*Barch et al., 2013*). The copes images of rewarding and aversive outcomes provided by HCP dataset were used in the analysis. The effect size of each contrast was extracted for each subject and each region of interest.

## Impulsivity measures

Measures from two behavioral tests of impulsivity were used for each subject: the Delay Discounting task and the Flanker Inhibitory Control Task. Detailed descriptions of these tasks are available in (*Barch et al., 2013*). Briefly, the Delay discounting paradigm (*Estle et al., 2006*) was selected as the measure of self-regulation and impulsive choice, which describes the temporal discounting of monetary rewards. Subjects choose between a smaller immediate and a larger delayed reward. A discounting measure of area-under-the-curve (AUC; [*Myerson et al., 2001*]) was calculated based on participants' choices across a series of trials, ranging from 1 (no discounting) to 0 (maximum discounting) with larger values representing less impulsive decisions. Only the high monetary amount ($40,000) was used here considering that its AUC value is approximately uniformly distributed across all subjects. The Flanker Inhibitory Control Task from the NIH Toolbox (http://www.nihtoolbox.org) was selected as a measure of response inhibition. During the Flanker task (*Eriksen and Eriksen, 1974*), participants are required to indicate the left–right orientation of a centrally presented arrow while inhibiting their attention to the surrounding flanking arrows. The final scores took into account the accuracy and reaction time on both congruent (in the same direction) and incongruent (in the opposite direction) trials, which provides a measure of inhibitory control in the context of selective visual attention (*Zelazo et al., 2014*). Higher scores represent both higher accuracy levels and faster reaction times, and therefore better inhibitory control.

## Seed regions

A mask of substantia nigra was generated from a 7T MRI atlas of basal ganglia based on high-resolution MP2RAGE and FLASH scans (*Keuken and Forstmann, 2015*) available at https://www.nitrc.org/projects/atag/. The entire region of SN (*Figure 2A*) was extracted from the probabilistic atlas with a

threshold of 33% of the population (i.e. retaining voxels labeled as SN in at least 10 out of 30 subjects) yielding masks of volume equal to approximately 300 mm$^3$ in each hemisphere. The SN seed mask overlayed on HCP average brain MRI images is illustrated in *Figure 1—figure supplement 1*.

Other regions of interest (ROI) of brain areas involved in reward and salience processing were defined as follows. Ventral striatum and ventral medial prefrontal cortex (vmPFC) were defined by drawing a 6 mm sphere around the peak coordinates from a fMRI meta-analysis of subjective value (*Bartra et al., 2013*). Two salience-related areas, the dACC and anterior insular cortex, were defined by drawing a 6 mm sphere around the peak coordinates of the salience network identified from resting state fMRI (*Seeley et al., 2007*). The MNI coordinates of these regions of interest are listed in *Table 1* and ROI are shown in *Figure 1—figure supplement 2*. Here, we hypothesized that ventral striatum and vmPFC were part of a value-coding system (with greater activation to reward than punishment), while dACC and anterior insula were more involved in salience-coding (i.e. responding similarly to reward and punishment). Note that we use a definition of salience as an equal response to rewarding and aversive events, as used in monkey electrophysiology (*Matsumoto and Hikosaka, 2009*) and human fMRI (*Rutledge et al., 2010*).

## Connectivity-based parcellation of SN

A data-driven connectivity-based brain parcellation procedure was used (*Figure 1*) as described in *Fan et al. (2016)*. First, probabilistic tractography was applied by sampling 5000 streamlines at each voxel within the seed mask of SN. A target mask was constructed for each subject that includes all brain voxels (white or gray matter) connecting to the seed region. The whole-brain connectivity profile for each SN voxel was then saved as a connectivity map, where the intensity shows how many streamlines reach the target area and is therefore a measure of the connectivity strength between the seed and target. These connectivity maps were used to generate a connectivity matrix with each row representing the whole-brain connectivity profile of one seed voxel. Next, a correlation matrix was calculated as a measure of similarity between the connectivity profiles of each voxel pair (*Johansen-Berg et al., 2004*). Spectral clustering (*Shi and Malik, 2000*) was applied to the similarity matrix to identify clusters with distinct connectivity profiles. We applied this procedure separately for each subject and each hemisphere to generate a series of parcellation maps for all individuals at different resolutions (i.e. numbers of regions/parcels). Considering the small size of the SN, we chose cluster numbers ranging from 2 to 8 in each hemisphere and chose the most stable and consistent parcellation map (see below).

An additional group-parcellation procedure was applied to summarize the general pattern of parcellation across subjects. Specifically, a consensus matrix $S$ was defined based on each individual parcellation map, with each element $S_{ij} = 1$ if and only if voxel $v_i$ and voxel $v_j$ belong to the same cluster. Then, a group consensus matrix was generated by averaging the consensus matrices from all subjects. The final group parcellation map was generated by performing spectral clustering again on the group consensus matrix (*Fan et al., 2016*; *Zhang et al., 2015*).

The optimum parcellation solution (i.e. number of parcels) was determined by evaluating the reproducibility of parcellation maps through a split-half procedure. Specifically, we randomly split the entire group into two non-overlapping subgroups 100 times and generated the group parcellation maps for each subgroup separately. The consistency between each pair of parcellation maps was evaluated by different stability indices, including normalized mutual information (NMI) (*Zhang et al., 2015*), Dice coefficient (*Zhang et al., 2014*) and Cramer's V (*Fan et al., 2014*). The average indices among 100 samples were calculated to represent the stability of each parcellation. The suitable cluster number was then determined by searching for the local peaks in the stability curve. In addition, topological similarity of the parcellation solutions between the two hemispheres was also calculated as a measure of stability (*Fan et al., 2016*).

## Connectivity profile of each SN subdivision

Based on the obtained parcellation map of SN, we mapped the anatomical connectivity profiles of each subdivision by performing probabilistic tractography with 10,000 streamlines from each SN subdivision. The resulting connectivity maps were first normalized by the size of the seed region and total number of streamlines (i.e. 10,000) in order to generate the relative tracing strength from the seed to the rest of the brain. A threshold of 0.001 (i.e. 10 out of 10,000) was then used to remove

noise effects of fiber tracking. The resulting individual tractograms were combined to generate a population map of the major fiber projections for each SN subdivision. Another probabilistic threshold of 50% was applied to the population fiber-tract maps (i.e. at least half of subjects showing each retained fiber tract). This resulted in a group averaged tractogram for each subdivision of SN. Finally, a maximum probability map (MPM) of fiber tracts, which represents distinct components of fiber projections for each subdivision, was also generated based on the population fiber-tract maps. Specifically, a connectome mask was first generated for each subdivision by binarizing its group tractography map with connectivity probability at 0.01. Note that different probability thresholds do not change the organizational pattern, but only enlarge or shrink the coverage of major fiber tracts (*Figure 4—figure supplement 1*). Each voxel within the combined connectome mask was then classified according to the SN subdivisions with which it had the highest connectivity. This calculation of MPM on probabilistic tractography has been widely used in subdividing brain structures, including thalamus (*Behrens et al., 2003*), amygdala (*Saygin et al., 2011*) and striatum (*Cohen et al., 2009*). Here, we use this method to generate the organizational topography of fiber projections among SN subdivisions.

A quantitative representation of the connectivity profiles was also generated by calculating the connectivity fingerprints between each SN subdivision and each cortical/subcortical area. A recently published brain atlas based on anatomical connectivity profiles (*Fan et al., 2016*) was chosen to define the target areas, consisting of a fine-grained parcellation of frontal, parietal, temporal, occipital cortex, limbic areas, as well as striatum and thalamus. The relative connectivity strength between each SN subdivision and each brain parcel for an individual was calculated by dividing the streamline counts by the size of each SN subdivision and each target area, as well as the total number of streamlines from each SN subdivision (i.e. 10,000). Next, we scaled this tracing strength by dividing it with the total tracing strength from the entire SN seed, which yields, for each target region and SN subdivision, the proportion of SN-target projections starting from that subdivision relative to the other two subdivisions. These normalized connectivity values were used to estimate the connectivity fingerprints for each SN subdivision.

## Neural activity during gambling task and correlations with impulsivity measures

This dissociation of reward value and motivational salience assessed by simultaneous manipulation of appetitive and aversive outcomes has been used often in the animal and human literature (*Bissonette et al., 2014*; *Roesch and Olson, 2004*). Several studies have provided evidence for this dissociation in prefrontal and cingulate areas (*Kahnt et al., 2014*; *Litt et al., 2011*; *Roesch and Olson, 2004*), and in midbrain structures (*Cohen et al., 2012*; *Matsumoto and Hikosaka, 2009*; *Rigoli et al., 2016*). Here, based on the rewarding and aversive outcomes of the fMRI gambling task, two new contrasts were defined, that is, a value signal as the difference in BOLD response to rewarding and aversive outcomes, and a salience signal as the average response to reward and punishment.

Moreover, to examine the association between task-related BOLD activity and impulsivity measures, we performed partial least squares (PLS) analysis using the PLScmd Matlab toolbox (http://www.rotman-baycrest.on.ca/pls/, [*McIntosh et al., 1996*; *McIntosh and Lobaugh, 2004*; *Mišić et al., 2016*]), which identifies linear combinations of brain activity and behavioral measures that maximally covary with each other. Briefly, a singular value decomposition was first performed on the correlation matrix of brain and behavioral data. The brain data included the BOLD response of three SN subdivisions and four other regions (VS, vmPFC, dorsal ACC, insula) under the condition of either value- or salience-coding. The behavioral data consisted of the two behavioral measures of impulsivity. A set of mutually orthogonal latent variables (LVs) was identified, where the left and right singular vector weights correspond to an optimal combination of BOLD response and behaviors, respectively. The effect size of each component was estimated as the ratio of the squared singular value to the sum of squared singular values from the decomposition. The overall significance of each pattern was assessed by permutation tests, whereby subject labels are randomly permuted and a distribution of singular values is generated under the null hypothesis that there exists no relation between the brain and behavioral data. p-Values were estimated as the proportion of permuted singular values that exceed the original singular value. The reliability with which each region or behavior contributes to the multivariate pattern was estimated by bootstrap resampling: participants were

sampled with replacement to construct a sampling distribution for each weight. To identify weights that (a) make a large contribution to the multivariate pattern and (b) are stable across many resamplings, we estimated bootstrap ratios: singular vector weights divided by their bootstrap-estimated standard errors. If the bootstrap distribution is approximately unit normal, the bootstrap ratio is approximately equivalent to a z-score (*Efron and Tibshirani, 1986*). We therefore thresholded all weights at bootstrap ratios equal to 1.96, corresponding to a 95% confidence interval.

## Data Availability

The resulting maps from our study are available at https://www.neurovault.org/collections/2860/.

## Acknowledgements

This work was supported by funding from the Canadian Institutes for Health Research and the Natural Sciences and Engineering Research Council of Canada. We thank Daniel Margulies for valuable discussions and the editors and reviewers of this manuscript for many helpful suggestions.

# Additional information

### Funding

| Funder | Grant reference number | Author |
| --- | --- | --- |
| Canadian Institutes of Health Research | Foundation Scheme | Alain Dagher |
| Natural Sciences and Engineering Research Council of Canada | Discovery Grant | Alain Dagher |

The funders had no role in study design, data collection and interpretation, or the decision to submit the work for publication.

### Author contributions

Yu Zhang, Conceptualization, Data curation, Formal analysis, Validation, Visualization, Methodology, Writing—original draft, Writing—review and editing; Kevin Michel-Herve Larcher, Software, Formal analysis, Methodology; Bratislav Misic, Conceptualization, Formal analysis, Methodology, Writing—original draft, Writing—review and editing; Alain Dagher, Conceptualization, Resources, Data curation, Formal analysis, Supervision, Funding acquisition, Investigation, Methodology, Writing—original draft, Project administration, Writing—review and editing

### Author ORCIDs

Alain Dagher  https://orcid.org/0000-0002-0945-5779

### Ethics

Human subjects: The authors agreed to the Open Access Data Use Terms of the Human Connectome Project (Van Essen et al 2013). Informed consent from participating individuals was obtained by the Human Connectome Project investigators. The Montreal Neurological Institute Research Ethics Board approved the use of Human Connectome Project data in the present project.

### Decision letter and Author response

Decision letter https://doi.org/10.7554/eLife.26653.031
Author response https://doi.org/10.7554/eLife.26653.032

# Additional files

### Supplementary files

• Transparent reporting form

DOI: https://doi.org/10.7554/eLife.26653.026

### Major datasets

The following dataset was generated:

| Author(s) | Year | Dataset title | Dataset URL | Database, license, and accessibility information |
|-----------|------|---------------|-------------|--------------------------------------------------|
| Yu Zhang, Kevin Michel-Herve Larcher, Bratislav Misic, Alain Dagher | 2017 | Anatomical and functional organization of the human substantia nigra and its connections | https://www.neurovault.org/collections/2860/ | Publicly available at neurovault |

The following previously published dataset was used:

| Author(s) | Year | Dataset title | Dataset URL | Database, license, and accessibility information |
|-----------|------|---------------|-------------|--------------------------------------------------|
| Van Essen DC, Smith SM, Barch DM, Behrens, TEJ, Yacoub E, Ugurbil K, WU-Minn HCP Consortium | 2013 | Human Connectome Project | http://www.humanconnectome.org | Publicly available at http://www.humanconnectome.org |

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
