## [Decision Letter]

Thank you for submitting your article "Anatomical and functional organization of the human substantia nigra and its connections" for consideration by *eLife*. Your article has been favorably evaluated by David Van Essen (Senior Editor) and three reviewers, one of whom, Heidi Johansen-Berg (Reviewer #1), is a member of our Board of Reviewing Editors. The following individual involved in review of your submission has agreed to reveal their identity: Suzanne N Haber (Reviewer #2).

The reviewers have discussed the reviews with one another and the Reviewing Editor has drafted this decision to help you prepare a revised submission.

Summary:

This is a thorough and thoughtful study assessing the anatomy and functional specialisations of subregions of the human substantia nigra. The study has a number of strengths including use of a relatively large, high quality dataset, careful methodological steps such as assessing repeatability across samples and assessing clustering across a range of different cluster numbers. Overall the connectivity profiles of the SN are convincing and relatively consistent with the literature with some exceptions (see below). Indeed, given the small size of the SN, these results are impressive. However, there are a few issues that should be addressed.

Essential revisions:

1) Subsection “Seed regions”, last paragraph – why would an equal response to reward and punishment be more involved in salience-coding? Given that these are both value-coding, isn't is possible that the dACC and insula may not distinguish between these valences, but nonetheless are coding general value. Using a neutral, surprise stimulus might be a better way to evaluate salience. It would be helpful to clarify the motivation for the stimuli chosen.

2) Due to its relatively small size, identifying SN in MR images is challenging. The authors use a substantia nigra ROI obtained from a publicly available 7T MRI atlas and register this to every individual subject. It would be good for the authors to illustrate (maybe in a figure supplement), how well this approach works. Are registration errors due to algorithmic inaccuracies/individual anatomy differences small enough to ensure reliable identification of SN across all subjects in dMRI space?

3) Some of the connectivity profiles are surprising and not supported by the literature. This is likely due to the pitfalls of diffusion and/or seed placement. Nonetheless, the authors should recognize them in the Discussion and address what might account for these mismatches. Moreover, these connections are not as black and white as the authors suggest. Indeed, some of their data show this, but the text doesn't reflect it. Below are some examples.

The mSNc preferentially is connected to the striatum compared to the vSN and lSNc. Since all parts of the SNc project to the striatum, one would expect both the v and lSNc to have strong projections there. One likely explanation is that the majority of fibers from the SNC to the striatum travel medially, turning dorsally and curving around the STN as they travel rostrally to the striatum. Thus, a seed placed in the mSNc will capture a large group (but not all) striatal fibers, and would likely include those from lateral and more ventral areas.

The lSNc mainly connects to the somatomotor and dorsal attention networks. The somatomotor connections are consistent with the literature, but why would this region target the dorsal attentional network primarily. One might expect a mixture of connections from both the lSNc and the vSNc. Indeed, in Figure 5, there appears to be a contribution from the vSNc, which is greater than that from the mSNC, albeit less than lSNc.

Why would the mSNc preferentially connect to visual areas?

Why would the vSN connect to the GPe, while the mSNc connect to the GPi and VP? Again, this is likely due to the seed placement and the streamlines that are captured rather than a reflection of actual connections.

4) Statistics for FMRI – the aim of the FMRI is to 'determine the different functional roles of different SN subdivisions'. However, the statistical contrasts run are not appropriate to address this aim as no direct comparisons between SN subdivisions are made. For both the whole brain analysis and the ROI analysis significant differences are found for reward v punishment in the VS and vmPFC whereas no significant differences for reward vs punishment are found in anterior insula, dorsal ACC and dorsal striatum. The opposing pattern is found for salience. However, a difference in significance does not imply a significant difference.

Similarly, these results do not show anatomically specific encoding of reward/salience. To make this conclusion requires direct statistical comparison between the regions – i.e. to test for an interaction between reward and region in the ROI analysis – and/or between the conditions – e.g. to directly contrast reward and salience in the whole brain analysis. Such tests are essential for the conclusion in the Discussion that "we found a double dissociation between coding of value and salience within SN subdivisions and their projections".

5) Correlations between FMRI and behaviour:

These correlations should be corrected for multiple comparisons (number of regions x number of behavioural measures). The authors show that value activity in medial SNc and VS correlated with delay discounting but not with flanker. Again, a difference in statistical significance doesn't imply that these correlations are significantly different. If the authors wish to conclude that these brain areas are more strongly correlated with response inhibition than motor inhibition then this requires a direct statistical test to compare the two correlations (e.g. using Fishers test). The same is true for the opposite patterns for dACC etc.

6) The same points about multiple comparisons and comparisons between correlations can also be made for the connectivity strength section.

---

## [Author Response]

Summary:This is a thorough and thoughtful study assessing the anatomy and functional specialisations of subregions of the human substantia nigra. The study has a number of strengths including use of a relatively large, high quality dataset, careful methodological steps such as assessing repeatability across samples and assessing clustering across a range of different cluster numbers. Overall the connectivity profiles of the SN are convincing and relatively consistent with the literature with some exceptions (see below). Indeed, given the small size of the SN, these results are impressive. However, there are a few issues that should be addressed.

We have tried to make the paper more focused. There are three main findings in order of importance: (1) the new parecellation of human substantia nigra (SN) showing distinct anatomical projections to the rest of the brain; (2) the dissociation between value and salience coding in SN and (3) evidence that the value and salience systems play dissociable roles in decisional and motor impulsivity.

Essential revisions:1) Subsection “Seed regions”, last paragraph – why would an equal response to reward and punishment be more involved in salience-coding? Given that these are both value-coding, isn't is possible that the dACC and insula may not distinguish between these valences, but nonetheless are coding general value. Using a neutral, surprise stimulus might be a better way to evaluate salience. It would be helpful to clarify the motivation for the stimuli chosen.

Here we are using the terms ‘value’ and ‘salience’ based on the usage introduced by Hikosaka in monkeys (Matsumoto and Hikosaka, 2009) and Glimcher in humans (Rutledge et al., 2010). In the neuronal recordings of Hikosaka in monkeys, DA neurons said to encode “value” are those that respond with increased firing to rewards (juice) and reduced firing to punishments (airpuff). On the other hand, neurons said to encode motivational “salience” respond equally to juice and airpuffs i.e. appear to encode the absolute value of the stimulus (whether rewarding or aversive). Note that this interpretation has been questioned by Fiorillo (Fiorillo, 2013) and Schultz (Schultz et al., 2017), who argue that no dopamine neurons encode actual aversive value. They suggest that dopamine neurons have an initial positive short-latency response to all stimuli that depends on their intensity, but that the subsequent response always encodes true value (activation for rewards, suppression for punishments). (Events on this time scale are too fast for fMRI to resolve.)

The monkey literature is concerned with individual DA neurons. However, similar arguments can apply at the spatial resolution of fMRI. Glimcher and Rutledge dissociated “value”, specifically reward prediction error (RPE), and “salience” (as defined here) in humans using fMRI and monetary reward (Rutledge et al., 2010). They define value in the usual sense for monetary reward, as the signed monetary amount of the reward. In their work, RPE (value) is found to be encoded by BOLD signal in the ventral striatum, while salience is encoded in the insula. Here a salience signal is one that reflects the absolute value of the monetary outcome (i.e. greater for large gains *and* large losses).

We now describe this as follows (Material and methods section “Neural activity during gambling task and correlations with impulsivity measures”):

“This dissociation of reward value and motivational salience assessed by simultaneous manipulation of appetitive and aversive outcomes has been used often in the animal and human literature (Roesch and Olson, 2004; Bissonette et al., 2014). […] Here, based on the rewarding and aversive outcomes of the fMRI gambling task, two new contrasts were defined, i.e. a value signal as the difference in BOLD response to rewarding and aversive outcomes, and a salience signal as the average response to reward and punishment.”

And in the Discussion:

“Note, however, that our inability to assign activation or connectivity to dopamine neurons (rather than SN neurons generally) means that our results do not contradict the alternative interpretation that no dopamine neurons encode actual aversive value (Fiorillo, 2013; Schultz et al., 2017). These authors suggest that dopamine neurons have an initial positive short-latency response to all stimuli that depends on their intensity, but that the subsequent response always encodes true value (activation for rewards, suppression for punishments).”

We agree with the reviewer(s) that salience could also be defined relative to outcome probability (i.e. salience equals surprise), but this was not manipulated in the current study. Given that HCP is an open access database not designed by us, we took the data available. Thus, we defined value encoding as greater BOLD response to rewarding than aversive outcomes, and salience encoding as greater response to rewards and punishments than neutral.

In this study, we used the cope images of rewarding and aversive outcomes provide by the HCP dataset. Two contrasts were defined, namely the value effect as the difference in BOLD response to rewarding and aversive outcomes, and the salience effect as the average response to reward and punishment. The effect size of each contrast was extracted for each subject and each region of interest.

We now clarify our use of the terms value and salience in the manuscript (Materials and methods, section “Seed regions”):

“Note that we use a definition of salience as an equal response to rewarding and aversive events, as used in monkey electrophysiology (Matsumoto and Hikosaka, 2009) and human fMRI (Rutledge et al., 2010).”

2) Due to its relatively small size, identifying SN in MR images is challenging. The authors use a substantia nigra ROI obtained from a publicly available 7T MRI atlas and register this to every individual subject. It would be good for the authors to illustrate (maybe in a figure supplement), how well this approach works. Are registration errors due to algorithmic inaccuracies/individual anatomy differences small enough to ensure reliable identification of SN across all subjects in dMRI space?

We performed quality control (visual inspection) on every registration of individual brains (T1w) to stereotaxic space. We show the location of the SN mask on the group averaged structural images and FA maps of 430 HCP subjects in stereotaxic space (Figure 1—figure supplement 1). Despite variations in brain shapes and gross anatomy, the approach identifies the correct location for human substantia nigra on average. All individual brains and the transformed SN masks can be provided if needed. The mask position is also compared to the Keuken and Forstmann (2015) atlas (Keuken and Forstmann, 2015) in the coronal plane in Author response image 1.

**Author response image 1. respfig1:** T atlas of Basal ganglia illustrating the location of Substantia Nigra. Figure adapted from the Keuken and Forstmann (2015) atlas.

Also, this mask is only used to limit the search volume for parcellation. Slight inaccuracies on the edges for certain individuals should not affect the final group parcellation, as it is based on the entire sample of 60 individuals. Finally, as a test of robustness, the parcellation was confirmed in a separate group of 60 individuals.

3) Some of the connectivity profiles are surprising and not supported by the literature. This is likely due to the pitfalls of diffusion and/or seed placement. Nonetheless, the authors should recognize them in the Discussion and address what might account for these mismatches. Moreover, these connections are not as black and white as the authors suggest. Indeed, some of their data show this, but the text doesn't reflect it. Below are some examples.

We have expanded the discussion for the connectivity profiles of SN subregions. First, it is true that there are large areas of overlap among projections from the three SN subdivisions, especially in prefrontal and striatal areas. Therefore, it is true that some of the figures could give the impression that connectivity was more segregated than it is in reality – however, other figures did show the overlap, especially among cortical projections of the three subdivisions (e.g. Figure 5).

In Figure 3, we used a relatively high threshold on tractography maps to emphasize the differences of cortical projections among SN subdivisions (probability of connectivity=0.05). With a lower threshold of connectivity probability at 0.01, we can now see the overlap among the major fiber tracts. We have added this as a supplemental figure (Figure 3—figure supplement 1).

Second, a winner-takes-all approach was used to generate the maximum probability map (MPM) of tractograms in Figure 4. This emphasizes the differences in connectivity profiles among SN subdivisions, but obscures areas of overlap. The method is explained in detail in the subsection “Subdivisions of SN”. Such an approach has been used to show distinct projections within subareas of thalamus (Behrens et al., 2003), amygdala (Saygin et al., 2011) and striatum (Cohen et al., 2009). Nonetheless, the MPM of fiber tracts only indicates the distinctive parts of SN projections, which depends on the chosen threshold of connectivity probability, as shown in Figure 4—figure supplement 1. The overlaps of fiber projections among SN subregions are now also shown in both the connectivity patterns (Figure 3—figure supplement 1 and Figure 4—figure supplement 2) and fingerprints (Figure 5).

We have added the following caveat to the Discussion:

“It is worth mentioning that this organizational pattern of SN projections (Figure 4) was detected by using a winner-takes-all approach, which emphasizes the distinct connections among subdivisions. […] For example, while SN-vmPFC connections were mostly derived from medial SNc, the other two subdivisions also made considerable contributions. This is consistent with the view that vmPFC is a connectional hub that integrates broad domains of information to support the valuation process during decision-making (Roy et al., 2012; Benoit et al., 2014).”

The mSNc preferentially is connected to the striatum compared to the vSN and lSNc. Since all parts of the SNc project to the striatum, one would expect both the v and lSNc to have strong projections there. One likely explanation is that the majority of fibers from the SNC to the striatum travel medially, turning dorsally and curving around the STN as they travel rostrally to the striatum. Thus, a seed placed in the mSNc will capture a large group (but not all) striatal fibers, and would likely include those from lateral and more ventral areas.

The statement on mSNc preferential connection to striatum was incorrect, and in contradiction to the rest of the paper. It was present in the legends to Figure 5 and Figure 5—figure supplement 1. Indeed, in the manuscript, we emphasized repeatedly in the text and figures that all three SN subregions connect to different parts of striatum (as shown in Figure 4 and Figure 4—figure supplement 2). We have changed the text of legends for Figure 5 and Figure 5—figure supplement 1.

Re: the potential medial-lateral bias of tractography results:

In Figure 4, dorsal medial SNc (colored in green) preferentially connects with ventral striatum, while ventral SN (colored in blue) mainly connects with dorsal striatum including body of Caudate and anterior putamen. This inverted dorsal-ventral arrangement is unlikely due to the propensity of diffusion imaging to identify tracts that are more direct. A medial-lateral gradient was also present such that medial SNc mainly connects with the medial part of Nuclear Accumbens and Globus Pallidus, while lateral SNc predominantly connects with lateral striatum, including tail of Caudate and posterior putamen. This medial-lateral organization might indeed be a result of the greater ease in identifying white matter connections that are shorter and less curved (Jbabdi et al., 2015). We now discuss this potential bias in the limitations section.

“Most of our results are unlikely to be due to such biases. […] Indeed the topological arrangement of SN-cortical connections exhibits a rotation from medio-lateral in SN to ventro-dorsal (and rostro-caudal) in cortex (Figure 3), as described previously for tractography of the ventral prefrontal cortex (Jbabdi et al., 2013).

The lSNc mainly connects to the somatomotor and dorsal attention networks. The somatomotor connections are consistent with the literature, but why would this region target the dorsal attentional network primarily. One might expect a mixture of connections from both the lSNc and the vSNc. Indeed, in Figure 5, there appears to be a contribution from the vSNc, which is greater than that from the mSNC, albeit less than lSNc.

The dorsal attentional network based on resting-sate functional MRI includes frontal eye fields (FEF) and precentral regions as well areas in intraparietal sulcus and superior partial lobe (Yeo et al., 2011). Yeo et al., illustrates the location of this dorsal attention network in both surface and multi-slice view.

All three subdivisions connect with this network (Figure 5—figure supplement 2), but the lSNc predominates. The connectional profiles and fingerprints (Figure 3 and Figure 5) showed that lSNc most strongly connects with the precentral gyrus and parietal areas, areas with well-described motor function. (Put another way, the resting state network called “dorsal attention” includes motor areas such as premotor cortex and posterior parietal cortex.)

Why would the mSNc preferentially connect to visual areas?

We agree: direct monosynaptic projection from SN to visual areas appear to be relatively sparse (Watabe-Uchida et al., 2012). Comparing our mSNc projections with the Brainstem tractographic atlas based on this dataset (Meola et al., 2016), we found some overlap between mSNc projections and the pathways of the temporo-parieto-occipito-pontine tract (TPO-PT) in the posterior limb of the internal capsule. The connection could therefore be artifactual. (Interestingly, however, one of the frequent symptoms of dopaminergic overdose in Parkinson’s disease is visual hallucinations, which is thought to implicate dopamine signaling in visual areas.)

We have added the following text as a qualification:

“On the other hand, some detected connections of SN were unexpected. For instance, the midbrain projections to visual cortex are sparse in rodents (Watabe-Uchida et al., 2012), but we detected relatively strong anatomical connections between mSNc and visual areas using both probabilistic and deterministic tractography (Figure 3—figure supplement 1 and Figure 3—figure supplement 2). This might be caused by an intersection of mSNc outflow with the temporo-parieto-occipito-pontine tract in the posterior limb of the internal capsule (Meola et al., 2016).”

Why would the vSN connect to the GPe, while the mSNc connect to the GPi and VP? Again, this is likely due to the seed placement and the streamlines that are captured rather than a reflection of actual connections.

Indeed, SN projections to pallidum showed a medial-lateral gradient (Figure 4), with medial SNc showing dominant connections with the internal part of the globus pallidus and ventral pallidum while vSN more strongly connected to external part of the globus pallidus. We agree that this could be due to a bias of tractography, and we now address this specifically within the new section on limitations:

“[…] it is quite possible that SN-pallidal connections described here are artifactual, as they do not accord with previously documented anatomical connections of basal ganglia.”

4) Statistics for FMRI – the aim of the FMRI is to 'determine the different functional roles of different SN subdivisions'. However, the statistical contrasts run are not appropriate to address this aim as no direct comparisons between SN subdivisions are made. For both the whole brain analysis and the ROI analysis significant differences are found for reward v punishment in the VS and vmPFC whereas no significant differences for reward vs punishment are found in anterior insula, dorsal ACC and dorsal striatum. The opposing pattern is found for salience. However, a difference in significance does not imply a significant difference.Similarly, these results do not show anatomically specific encoding of reward/salience. To make this conclusion requires direct statistical comparison between the regions – i.e. to test for an interaction between reward and region in the ROI analysis – and/or between the conditions – e.g. to directly contrast reward and salience in the whole brain analysis. Such tests are essential for the conclusion in the Discussion that "we found a double dissociation between coding of value and salience within SN subdivisions and their projections".

We agree. We have performed more statistical analysis to address this issue.

First, a whole-brain contrast between value- and salience-related BOLD responses confirmed that there were separated brain systems for value and salience coding. The value-related response was defined as the difference in BOLD response to winning versus losing outcomes, while the salience-related response was defined as the average response to wins and losses. A paired t-test was used to show the differences in brain response to value vs salience (thus defined). For instance, ventral striatum (peak: x=0, y=6, z=-12; T=27.76) and vmPFC (peak: x=0, y=58, z=-4; T=30.03) showed greater evidence for value-coding, while dACC (peak: x=-4, y=8, z=26; T=11.91) and anterior insula (peak: x=40, y=20, z=-2; T=33.20) showed greater evidence for salience-coding.

Second, “two-way repeated measures ANOVA revealed a significant interaction effect between SN subdivisions and BOLD response to reward or punishment (F=6.6, p=0.0014). As shown in Figure 6, all three SN subregions were activated by both reward and punishment, but only medial SNc showed significantly greater neural activity to monetary gains than losses (T= 3.96, p< 0.0001). Moreover, among the subdivisions of SN, medial SNc showed significantly higher BOLD response to positive value (gains) compared to the other two subdivisions (T= 3.52, p=0.0005 for vSN; T= 2.98, p=0.003 for lateral SNc), as shown in Figure 6—figure supplement 3.”

“A significant interaction effect was also detected between the four target regions (VS, vmPFC, dACC, insula) and their BOLD response to reward or punishment (F=30.26, p-value=3e-19).”

5) Correlations between FMRI and behaviour:These correlations should be corrected for multiple comparisons (number of regions x number of behavioural measures). The authors show that value activity in medial SNc and VS correlated with delay discounting but not with flanker. Again, a difference in statistical significance doesn't imply that these correlations are significantly different. If the authors wish to conclude that these brain areas are more strongly correlated with response inhibition than motor inhibition then this requires a direct statistical test to compare the two correlations (e.g. using Fishers test). The same is true for the opposite patterns for dACC etc.

We agree. We now present the results corrected for multiple comparisons and compare the two correlations using Fisher’s test (Figure 7—figure supplement 1). However, we start with a more sophisticated statistical analysis using partial least squares (PLS), which identifies orthogonal latent variables to show the dissociation between decisional and motor impulsivity. The PLS approach is described in the Materials and methods section:

“Moreover, to examine the association between task-related BOLD activity and impulsivity measures, we performed partial least squares (PLS) analysis using the PLScmd Matlab toolbox (http://www.rotman-baycrest.on.ca/pls/, (McIntosh et al., 1996; McIntosh and Lobaugh, 2004; Mišić et al., 2016)), which identifies linear combinations of brain activity and behavioural measures that maximally covary with each other. […] We therefore thresholded all weights at bootstrap ratios equal to 1.96, corresponding to a 95% confidence interval.”

The result of this analysis supports the anatomical dissociation between decisional and motor impulsivity:

“To characterize the relationship between BOLD activity and impulsivity measures, behavioral PLS analysis was performed. […] The resulting brain networks and behavioral measures are consequently independent if they belong to different components.”

Finally, we also provide the individual correlations between BOLD and impulsivity (as before) including Bonferroni correction and Fisher’s test:

“Individual correlation analysis between BOLD response and impulsivity measure was also applied to confirm the observed brain-behaviour relations (Figure 7—figure supplement 1). […] Moreover, the correlation of value-coding response in VS was significantly higher with Delay-Discounting measures than inhibitory control scores (p=0.02), while the association of salience-coding response in anterior insula was significantly stronger with inhibitory control than Delay-Discounting (p=0.008).”

6) The same points about multiple comparisons and comparisons between correlations can also be made for the connectivity strength section.

We agree. We have decided to remove this part of the analysis (and Figure 8) from the paper. The relationships between tractography findings and impulsivity, while intriguing, did not survive multiple comparisons and in any case were a minor part of the paper.